# Are Physical Therapeutics Important for Stroke Patients to Recover Their Cardiorespiratory Fitness?

**DOI:** 10.3390/medicina57111182

**Published:** 2021-10-31

**Authors:** Ki-Bok Choi, Sung-Hyoun Cho

**Affiliations:** 1Team of Rehabilitation Treatment, Chosun University Hospital, 365, Pilmun-daero, Dong-gu, Gwangju 61453, Korea; csh2007134@chosun.ac.kr; 2Department of Physical Therapy, Nambu University, 23 Cheomdan Jungang-ro, Gwangsan-gu, Gwangju 62271, Korea

**Keywords:** cardiorespiratory fitness, Delphi technique, therapeutics, stroke, rehabilitation

## Abstract

*Background and Objectives*: Aspects of improving cardiorespiratory fitness should be factored into therapeutics for recovery of movement in stroke patients. This study aimed to recommend optimized cardiorespiratory fitness therapeutics that can be prescribed to stroke patients based on a literature review and an expert-modified Delphi technique. *Materials and Methods*: we searched PubMed, Embase, CINAHL, and Cochrane databases and yielded 13,498 articles published from 2010 to 2019 to support the development of drafts. After applying the exclusion criteria, 29 documents were analyzed (drafts, 17 articles; modified Delphi techniques, 12 articles). This literature was reviewed in combination with the results of a modified Delphi technique presented to experts in the physical medicine and rehabilitation field. Analysis of the literature and survey results was conducted at the participating university hospital. *Results*: the results of this analysis were as follows: first, 12 intervention items derived through a researcher’s literature review and a Delphi technique questionnaire were constructed using the Likert scale; second, we asked the experts to create two modified Delphi techniques by reconstructing the items after statistical analysis for each order comprising five categories, and 15 items were finally confirmed. *Conclusions*: the recommendations in this study may lead to the development of a standard decision-making process for physiotherapists to improve their patients’ cardiorespiratory fitness. Moreover, the study results can help prescribers document patient care to reduce prescription errors and improve safety. In the future, multidisciplinary studies could potentially provide better therapeutics alternatives for cardiorespiratory fitness.

## 1. Introduction

Improved cardiopulmonary health reduces all-cause mortality [1,2]. Stroke and cardiovascular diseases can cause death due to low cardiopulmonary function [3]. Stroke can decrease motor function through neurological damage and limited physical movement, resulting in low physical activity [4]. Low cardiopulmonary health and, in particular, lack of physical activity are modifiable risk factors for stroke and are associated with greater illness severity and worse functioning [1,2]. The initial symptoms of severe stroke can lead to the patient becoming sedentary or spending most of their time bedridden [5]. Low physical activity results in vascular impairments in chronic stroke patients [6], such as reduced lung diffusion capacity due to the active use of partially paralyzed muscles, chest wall and diaphragmatic movement, ventilation-perfusion inconsistency, or partial respiratory failure due to reduced lung volume [7,8,9].

Expiratory dysfunction is characterized by a decrease in the movement of the paralytic diaphragm, intercostal muscles, and abdominal muscles [10]. A decrease in chest wall movement can promote secondary muscle fibrosis in the ribs, further limiting inspiration owing to decreased maximum inspiration pressure [11,12]. This is related to the poor cardiovascular health of stroke patients and is explained by the difference between the measured ventilation volume after exercise and the maximum ventilation volume (reduced ventilatory reserve) [13,14]. The maximum oxygen intake (VO2max) decreases to 10–17 mL/kg/min within 30 days of stroke episode and may not rise above 20 mL/kg/min after 6 months [15], resulting in the maximum oxygen intake of a stroke patient being 25–45% lower than that in healthy participants [16].

Stroke patients have decreased muscle function and reduced exercise tolerance; therefore, it is crucial to improve cardiopulmonary function with aerobic exercise and activities that strengthen muscles, such as the intercostal muscle and diaphragm responsible for inspiration and expiration [17,18]. Aerobic exercise can improve cardiopulmonary function, gait [19], and blood flow in leg injuries [20]. Cardiopulmonary function can be improved in stroke patients using methods developed to improve cardiac breathing [21,22]. Consequently, exercise for stroke patients provides a strong stimulus that promotes cardiovascular responses sufficient to meet muscle metabolic needs for normal activity [23]. Aerobic exercises do not simply refer to just one exercise for better cardiopulmonary functioning; it is crucial to improve cardiopulmonary functioning by applying these exercises through various methods.

Post-stroke treatment is complex and associated with long-term physiological and psychological, including motor and functional, balance, cognitive, and emotional problems [24,25]. In recent years, multidisciplinary teams have been formed that include health professionals such as doctors, physiotherapists, nurses, occupational therapists, speech therapists, psychologists, and nutritionists from various disciplines, with patients and caregivers also important members of the team [26,27]. Various treatments such as psychotherapy, physiotherapy, exercise therapy, speech therapy, and occupational therapy are employed in the rehabilitation process of stroke patients [28,29,30,31]. To improve cardiorespiratory function in stroke patients, aerobic exercises such as those that strengthen the respiratory muscles, task-oriented exercises, inspiratory muscle resistance exercises, and climbing stairs have been actively investigated [32,33,34,35,36,37]. However, there is still a lack of awareness about cardiorespiratory physiotherapy for stroke patients, and no studies have proposed a standardized guideline to implement cardiorespiratory physiotherapy interventions.

This study aimed to collate a set of cardiorespiratory physiotherapy measures for stroke patients using a modified Delphi technique to achieve consensus between the literature and an expert panel. The modified Delphi technique is not a structured response method by a panel of experts, such as a mandatory agreement. Instead, it modifies the existing Delphi technique to explain the opinions of our expert panel based on a structured questionnaire developed by the researchers [38]. A Delphi technique used to formulate healthcare indicators reports that 49/78 (68%) of the panels use a modified Delphi technique [39].

We examined data from previous studies that explored cardiorespiratory fitness interventions for stroke patients and used a modified Delphi technique with experts’ opinions to develop clinically applicable guidelines for establishing interventions for cardiorespiratory physiotherapy in stroke patients.

## 2. Materials and Methods

### 2.1. Literature Review and Study Selection

We systematically searched the literature published between 1 January 2010, and 30 June 2019. Medical Subject Headings (MeSH), Emtree, and CINAHL subject headings were used in the search strategy. We conducted our searches using PubMed, EMBASE, CINAHL Plus with Full Text, and Cochrane Library databases (Table 1 and Table 2). Two authors conducted the literature search and the study inclusion processes (Table 3). In cases where a consensus was not reached, it was subsequently resolved by a third-party expert.

### 2.2. Modified Delphi Methodology

The panel of experts who agreed to participate in the study consisted of 9 licensed physiotherapists with at least 10 years of clinical or research experience in cardiorespiratory physiotherapy. From 15 July 2019 to 31 August 2019, drafts of the questionnaires using Likert scales and free text responses were sent to experts, and consultation for data and content collection was conducted in person or via e-mail (Appendix A). The panel consisted of experts on cardiorespiratory physiotherapy that agreed to participate in the study (first trial, 9; second trial, 9).

### 2.3. Principal Component Analysis

All data were statistically analyzed using IBM SPSS Statistics for Windows/Macintosh, version 20.0 (IBM Corp., Armonk, NY, USA). Descriptive statistical analyses were conducted to confirm the general characteristics of the panel and to verify their understanding of the intervention items. Based on the displayed fitness values of the Likert scale, content validity and reliability were analyzed, and validity (convergence and consensus) and stability were calculated to revise and supplement the opinions of the expert panelists.

## 3. Results

### 3.1. Data Extraction Criteria

The final dataset to be analyzed was selected through a systematic screening of the collected articles using Endnote software (X9.1, Clarivate Analytics, Philadelphia, PA, USA). In order to perform this scoping review, the PRISMA-ScR model was followed [40], and the 17 selected articles were analyzed for draft development (Figure 1).

### 3.2. The Delphi Process

In the first modified Delphi technique, the mean goodness of fit of the intervention items was 4.07 ± 0.30 points based on the Likert scale responses. Internal consistency reliability was set at α = 0.312 for the intervention items.

The second modified Delphi technique was conducted with the same panel of 9 experts, and the collection rate was 100%. The mean goodness of fit of the intervention items was 4.66 ± 0.23 points based on the Likert scale responses. For the content validity ratio (CVR), all other items exceeded the minimum score. Likewise, the mean convergence was 0.35, the mean consensus improved to 0.86, and the mean stability was 0.11 (Table 4). The accuracy of the statements for each item and the item validity all exceeded the minimum score. Patient and caregiver education (aerobic exercise, self-assisted coughing) were added by expert panelists.

### 3.3. Deriving Final Evaluation Items

The results of the first modified Delphi technique showed that diagonal pattern gymnastics motion with sitting, high-intensity interval training, aquatic treadmill exercise, feedback-controlled robotics-assisted treadmill exercise, and circuit aerobic exercise scored below the minimum values and were subsequently deleted, as the internal consistency reliability was very low (α = 0.312). By contrast, in the second modified Delphi technique, none of the items were deleted when the content validity value was less than the minimum value, and the internal consistency reliability was extremely high (α = 0.715).

Fifteen optimal cardiorespiratory physiotherapy interventions for use with stroke patients were derived based on the literature review in this study, which was then reorganized through modified Delphi techniques by the expert panelists (Table 5).

## 4. Discussion

The purpose of this study was to present selective recommendations from expert panelists for interventional methods to improve the quality of cardiorespiratory physiotherapy and use them in clinical practice. In hospital-supported institutions, intervention methods are determined and implemented based on individual patient evaluation information [41]. However, there are currently no universal guidelines for interventional methods related to cardiorespiratory physiotherapy for stroke patients.

According to previous studies, if the panelists’ consensus was >0.75 points and the convergence was 0.50 points, the panel’s opinions were judged to find the points of agreement [42,43]. In this study, the mean convergence was 0.58, while the mean consensus and mean stability were 0.69 and 0.21, respectively. The opinion gathering and consensus among expert panelists were not adequate. The average stability was 0.21 points, and it was found that no additional questionnaire was required. However, the convergence and consensus of the expert panelists were not consistent. The need for expert panel responses to open-ended questions was raised, leading to the implementation of a second modified Delphi technique.

After employing the first modified Delphi technique, the open panel opinions of expert panelists were as follows. First, the main subject area for the items was the expansion of the airway and ventilation area, strength exercise area for respiratory rehabilitation, strength exercise area for cardiac rehabilitation, cardiopulmonary endurance area, and education (caregiver education). Second, although it is ideal to suggest an intervention method through a multidisciplinary approach [44,45], it is difficult to seek cooperation from various medical experts, and there are practical difficulties, such as considerable research time and cost. Third, interventions for lower-extremity ergometer exercise and aerobic exercise were added based on previous evidence [46,47]. Fourth, as proposed by the expert panel, the caregiver education items were subdivided into aerobic exercise and self-assisted cough. Eight items were newly formulated by the expert panel, including segmental breathing, cervical, and thoracic mobilization [48], active-assisted stretching exercise [49], IMT [50,51], resistance training [52,53], L/E ergometer exercise [47], aerobic exercise, principles of FITT [46], and caregiver education (aerobic exercise and self-assisted coughing). The reason for a large number of additional items after the first modified Delphi technique was that no consensus was reached by the expert panelists.

After implementing the second modified Delphi technique, the content validity value exceeded the minimum value; therefore, no items were deleted. The reliability of all items (internal consistency of items) was high (α = 0.715). The average convergence was 0.35 points, the average consensus was 0.86 points, and the average stability level was 0.11 points. Among the intervention items, caregiver education was revised to (regular) patient and caregiver education, and detailed information on cardiac breathing-related symptom education, coping education according to the manifestation of heart-related symptoms, exercise intensity training in a range in which symptoms do not appear, aerobic exercise, and self-assisted coughing was added. For a treadmill walk exercise at 3 km/h, the energy expenditure was 2.43 metabolic equivalents (METs), and when the slope was increased by 10% at the same speed to achieve higher exercise intensity, the energy expenditure was 5 METs. However, on a flat slope, even if the treadmill speed was increased two-fold to 6 km/h, the energy expenditure was only 3.85 METs, suggesting that it was more beneficial to stroke patients if the exercise intensity was increased by increasing the degree of the slope rather than by increasing the speed [54]. Recent studies have suggested that high-intensity exercises, including high-speed power training, can be safe, new intervention methods for post-stroke cardiopulmonary rehabilitation [55,56]. In addition, it may be more effective to selectively utilize lower-extremity ergometer exercises in the clinic, depending on the risk of falls and the patient’s condition.

The results of this study show that the 15 selected interventions for stroke patients are effective strategies to improve cardiorespiratory fitness caused by VO2max, unilateral diaphragm, and respiratory muscle paralysis. This finding can be clinically important, as VO2max can be reduced by 25–45% in stroke patients and can cause significant problems with a respiratory capacity [57]. Collectively, these findings may support recommendations for intervention methods to prevent or mitigate the reduction of VO2max, or even to restore damaged VO2max, unilateral diaphragm, and respiratory muscles. 

No commercial funding was received to conduct this study, thus strengthening the validity of our recommendations. However, this study also has a few limitations. First, only a limited number of expert panels and researchers influence the search strategy utilized as well as the value and accessibility of recognized research on this topic. In addition, we selected panelists with contrasting experience in the field of physiotherapy and cardiopulmonary. In the future, we plan to accumulate more evidence by examining studies related to cardiorespiratory physiotherapy for stroke patients.

## 5. Conclusions

Research using questionnaire methods for experts utilizes the Delphi process, which can be performed quickly in clinical practice. It is worth noting that e-mailing the way expert intervention works to take advantage of the non-face-to-face benefits has resulted in a modified Delphi process. This study has established a fundamental system to apply cardiorespiratory physiotherapy interventions to stroke patients by systematic reviews and modified Delphi technique results.

In clinical practice, cardiorespiratory fitness programs are required in intervention plans for recovery of motor function in stroke patients, and for this, the final 15 items derived from this study can be applied. In addition, the derived intervention items can be selectively applied according to the individual health status of stroke patients; therefore, customized interventions are possible.

## Figures and Tables

**Figure 1 medicina-57-01182-f001:**
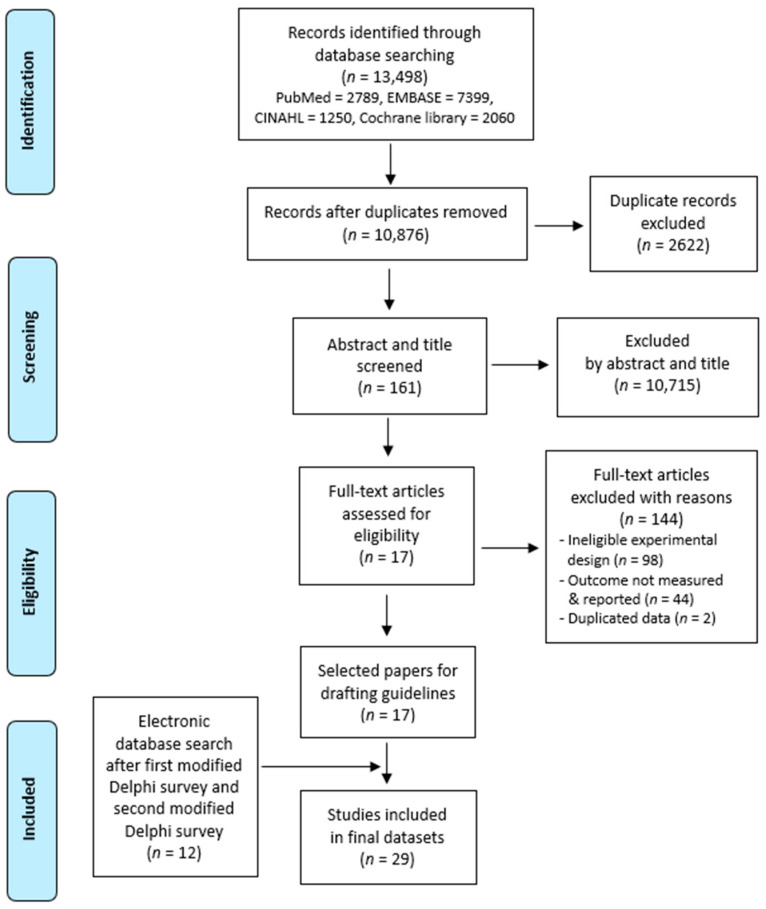
Flow diagram showing the identification of relevant literature in the PRISMA-ScR format.

**Table 1 medicina-57-01182-t001:** MeSH search terms.

User Group	Field of Health Research	
Stroke	Cardiac rehabilitation	Exercise
Cerebrum	Respiratory therapy	Heart
Brain	Patient outcome assessment	Rehabilitation
	Cardiorespiratory fitness	

**Table 2 medicina-57-01182-t002:** Emtree search terms.

User Group	Field of Health Research	
Stroke patient	Cardiorespiratory fitness	Exercise
Cerebrovascular disease	Respiratory care	Heart rehabilitation
Brain	Breathing	Breathing exercise
	Lung function test	Outcome assessment
	Checklist	Respiratory tract parameters

**Table 3 medicina-57-01182-t003:** Define inclusion criteria.

Participants	Adult stroke patients aged > 18 years
Intervention	A single intervention (single intervention group) or a combination of two or more interventions (combination group)
Comparisons	Groups receiving no interventions for cardiorespiratory physiotherapy or those receiving typical physiotherapy without interventions
Outcomes	A single variable or two or more variables containing exercise interventions related to physiotherapy of cardiac and respiratory functions
Study design	Observational descriptive study, case-series, observational analytical study, consensus document, editorial, cross-sectional study, case-control study, cohort study, systematic literature reviews, systematic review, meta-analysis, randomized controlled trial (RCT), quasi-experimental design

**Table 4 medicina-57-01182-t004:** Results of items after the second modified Delphi technique.

Domain	No	Intervention Item	Mean	SD	CVR	Q1	Median	Q3	Stability	Convergence	Consensus
Airway expansion and ventilation	1	Air stacking exercise or manual hyperinflation	4.56	0.53	1.00	4.00	5.00	5.00	0.12	0.50	0.80
	2	Manually assisted cough	4.89	0.33	1.00	5.00	5.00	5.00	0.07	0.00	1.00
	3	Mechanical insufflator-exsufflator	4.67	0.50	1.00	4.00	5.00	5.00	0.11	0.50	0.80
	4	Noninvasive intermittent positive pressure ventilation	4.56	0.53	1.00	4.00	5.00	5.00	0.12	0.50	0.80
	5	Segmental breathing	4.56	0.53	1.00	4.00	5.00	5.00	0.12	0.50	0.80
	6	Chest expansion resistance exercise	4.78	0.44	1.00	4.50	5.00	5.00	0.09	0.25	0.90
	7	Cervical and thoracic mobilization	4.78	0.44	1.00	4.50	5.00	5.00	0.09	0.25	0.90
	8	Active assisted stretching exercise	4.44	0.73	0.78	4.00	5.00	5.00	0.16	0.50	0.80
Strength training for respiratory rehabilitation	9	Core exercise combined with abdominal breathing	4.56	0.53	1.00	4.00	5.00	5.00	0.12	0.50	0.80
	10	Inspiratory muscle training	4.44	0.73	1.00	4.00	4.00	5.00	0.12	0.50	0.75
Strength training for cardiac rehabilitation	11	Resistance training	4.44	0.73	0.78	4.00	5.00	5.00	0.16	0.50	0.80
Cardiopulmonary endurance	12	L/E ergometer exercise	4.67	0.50	1.00	4.00	5.00	5.00	0.11	0.50	0.80
	13	Treadmill exercise	4.78	0.44	1.00	4.50	5.00	5.00	0.09	0.25	0.90
	14	Aerobic exercise: principles of FITT	4.89	0.33	1.00	5.00	5.00	5.00	0.07	0.00	1.00
Education	15	Caregiver education (Aerobic exercise and self-assisted coughing)	4.89	0.33	1.00	5.00	5.00	5.00	0.07	0.00	1.00
Intervention		Average	4.66	0.23	0.97	4.30	4.93	5.00	0.11	0.35	0.86

**Table 5 medicina-57-01182-t005:** Cardiorespiratory physiotherapy final intervention items for stroke patients.

Domain	No	Final Items
Airway expansion and ventilation	1	Air stacking exercise or manual hyperinflation
	2	Manually assisted cough
	3	Mechanical insufflator-exsufflator
	4	Noninvasive intermittent positive pressure ventilation
	5	Segmental breathing
	6	Chest expansion exercise
	7	Cervical and thoracic mobilization
	8	Active assisted stretching exercise: four sections of neck, upper thoracic, pectoralis major, lateral chest
Strength training for respiratory rehabilitation	9	Core exercise combined with abdominal breathing
	10	Inspiratory muscle training
Strength exercises for cardiac rehabilitation	11	Resistance training
Cardiopulmonary endurance	12	L/E ergometer exercise
	13	Treadmill exercise
	14	Aerobic exercise: principles of FITT
Education	15	Patient and protector education (aerobic exercise and self-assisted coughing)

## Data Availability

All data relevant to the study are included in the article. Data were collected from studies published online or publicly available, and specific details related to the data will be made available upon request.

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
