# Peer review of "Are Physical Therapeutics Important for Stroke Patients to Recover Their Cardiorespiratory Fitness?"

_medicina, 2021, doi:10.3390/medicina57111182_

Round 1

Reviewer 1 Report

Rehabilitation after stroke is an important issue.  Items can be important in prescribed exercise. However, I still propose some fundamental changes.

  1. Change the words ‘papers’ to ‘articles’
  2. Add more study to the introductions on benefit rehabilitation in stroke patients
  3. Better information about stroke recovery and rehabilitation is needed.
  4. Strokes decrease motor functions; no one can

‘Strokes can decrease motor function due to neurological damage and limited physical movement, resulting in low physical activity’.

  1. Aerobic exercises are not only one exercise for better cardiopulmonary functions, it is crucial to improve cardiopulmonary function with aerobic exercise ‘
  2. Add another study on rehabilitation after stroke to the discussion
  3. Add a limitation to the study

Author Response

We attached the file.

Reviewer 2 Report

Dear Authors,

the topic of your paper is an important chapter for Stroke Rehabilitation but stroke and cardiovascular disease mortality is related to many factors and not only low cardiorespiratory fitness (Introduction 32-33-34). Infact stroke patients are very complex and the therapeutic approach needs multi-professional rehabilitation teams including several medical specialities and types of health professionals (neurologist, physiatrist, physical therapist, speech therapist, occupational therapist, psychologist, and social worker).  The Individual Rehabilitative Project is essential to coordinate the different rehabilitation professionals involved in patient care and this is not an “ideal approach” (Discussion 159-162) yet a well documented methodology, accepted in the clinical and research community. Thus in the introduction section (74-75) you declared unobtainable aims  with your approach  that includes only physical therapists and excludes others medical and  health-care professionals. Excellent Guidelines for stroke patients have been published using the GRADE method and are widely accepted. Unlike the introduction, in the discussion section (139-142) you stated more appropriate aims (“selective recommendations”).  In the title I suggest adding “physical “before therapeutics. Tables 4 have to be completely revised in an orderly fashion .

Despite these substantial issues, your article provides a very well done review of cardiovascular interventions supported by the modified Delphi technique that can positively impact the cardiorespiratory programs in the Individual Rehabilitative Project of many stroke patients, of course the timing and the intensity of these interventions have to be evaluated case by case. For these reasons I recommend a major revision of the paper. If well revised I think it would be of value when published.

Author Response

We attached the file.

Round 2

Reviewer 1 Report

It is limitation, no conclussion 

"There are a limited number of expert panels and researchers who can influence the 228 search strategy utilized and the value and accessibility of recognized research on this topic. 229 However, the panelists with contrasting experience in the field of physiotherapy and car-230 diopulmonary were selected. Further, no commercial funding was received for this study, 231 thus strengthening the recommendations obtained. In the future, we will continue to de-232 velop further evidence through various studies related to cardiorespiratory physiotherapy 233 for stroke patients." 

Author Response

We attached the file.
